# Large Multi-modality Model Assiste
# AI-Generated Image Quality Assessment

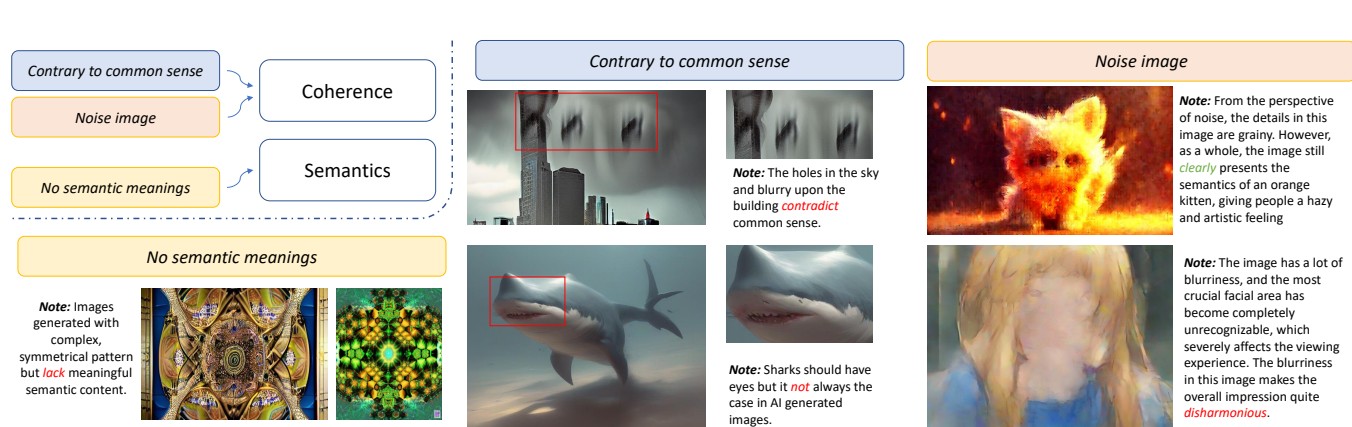

**Figure 1: The quality of AI-generated images is greatly influenced by the semantic content, i.e., the coherence and existence of semantic content. The *coherence* of a picture's semantic content is crucial for providing a logically sound visual experience for human viewers, and images lacking *semantic content* fail to effectively communicate their intended design, reducing viewer engagement and satisfaction. This article primarily focuses on "Contrary to common sense" (CCS), "Noise image" (NI) and "No semantic meanings" (NSM). We categorize "CCS" and "NI" under the coherence of semantics, while "NSM" falls under the existence of semantics.**

## ABSTRACT

Traditional deep neural network (DNN)-based image quality assessment (IQA) models leverage convolutional neural networks (CNN) or Transformer to learn the quality-aware feature representation, achieving commendable performance on natural scene images. However, when applied to AI-Generated images (AGIs), these DNN-based IQA models exhibit subpar performance. This situation is largely due to the semantic inaccuracies inherent in certain AGIs caused by uncontrollable nature of the generation process. Thus, the capability to discern semantic content becomes crucial for assessing the quality of AGIs. Traditional DNN-based IQA models, constrained by limited parameter complexity and training data, struggle to capture complex fine-grained semantic features, making it challenging to grasp the existence and coherence of semantic content of the entire image. To address the shortfall in semantic content perception of current IQA models, we introduce a large *M*ulti-modality model *A*ssisted *A*I-*G*enerated *I*mage *Q*uality *A*ssessment (**MA-AGIQA**) model, which utilizes semantically informed guidance to sense semantic information and extract semantic vectors

through carefully designed text prompts. Moreover, it employs a mixture of experts (MoE) structure to dynamically integrate the semantic information with the quality-aware features extracted by traditional DNN-based IQA models. Comprehensive experiments conducted on two AI-generated content datasets, AIGCQA-20k and AGIQA-3k show that MA-AGIQA achieves state-of-the-art performance, and demonstrate its superior generalization capabilities on assessing the quality of AGIs. The code will be available.

## CCS CONCEPTS

• **Computing methodologies → Computer vision tasks**.

## KEYWORDS

Image Quality Assessment, AI-Generated Image, Large Multi-modality Model, Mixture of Experts

## 1 INTRODUCTION

The rapid advancement of artificial intelligence (AI) has led to a proliferation of AI-generated images (AGIs) on the Internet. However, current AI-driven image generation systems often produce multiple images, necessitating manual selection by users to identify the best ones. This labor-intensive process is not only time-consuming but also a significant barrier to fully automating image processing pipelines. Visual quality, as an important factor to select attractive AGIs, has gained lots of attention in recent years [17, 20]. In this paper, we focus on how to evaluate the visual quality of AGIs, which on the one hand can be used to filter high-quality images

from generation systems and on the other hand, can sever as reward function to optimize image generation models [2], propelling progress in the field of AI-based image generation techniques.

While a substantial number of deep neural network (DNN)-based image quality assessment (IQA) models, such as HyperIQA [39], MANIQA [49], DBCNN [55], etc., have been developed, these models were specifically designed for and trained on natural scene images. When applied directly to AGIs, these models often exhibit poor performance. This is due to the fact that quality assessment of natural images primarily targets issues such as blur, noise, and other forms of degradation caused by photography equipment or techniques, which are not applicable to AGIs as they do not undergo such degradation during the generation process. Therefore, overemphasizing factors like blur or noise during the evaluation of AGIs is inappropriate.

As shown in Figure 1, AI-generated images, derived from advanced image generative models such as generative adversarial networks (GANs) [23], diffusion [10] and related variant [4, 6, 11, 29, 32–34, 48], often exhibit issues not commonly found in naturally captured images. Visual quality of AGIs depends not only on basic visual features such as noise, blur [18, 38, 58], etc., but also on more intricate semantic perception [17], such as existence of reasonable semantic content, scene plausibility, and the coherence among objects [19, 43, 44, 46, 57]. Although re-training existing IQA models on AGIs datasets leads to improved outcomes, it fails to achieve optimal performance. One reason is that traditional DNN models, especially early convolutional neural networks (CNNs), despite their notable achievements in tasks like image recognition and classification [9, 37, 41], still struggle to grasp the fine-grained semantic content of images [56]. What's more, traditional DNN-based IQA models fail to capture the intrinsic characteristics essential for assessing image quality and thus exhibit poor generalization abilities. Hence, we argue that the quality assessment models of AGIs are still in their infancy and need further exploration.

To address the issue of semantic awareness, we resort to large multi-modality models (LMMs). Because LMMs is typically pre-trained on large-scale datasets and has already learned a rich set of joint visual and language knowledge, it can effectively capture the fine-grained semantic features relevant to input prompts. However, LMMs perform excellently in high-level visual understanding tasks [1, 16], yet they do not perform well on tasks that are relatively simple for humans, such as identifying structural and textural distortions, color differences, and geometric transformations [47]. In contrast, traditional deep learning networks excel at perceiving low-dimensional features and can fit better to the data distribution of specific task [12]. Therefore, the idea of combining LMMs with traditional deep learning networks is a natural progression.

In this paper, we introduce a large **M**ulti-modality model **A**ssisted **A**I-**G**enerated **I**mage **Q**uality **A**ssessment (**MA-AGIQA**) framework, which enhances the capacity of traditional DNN-based IQA models to understand semantic content by incorporating LMM. Our approach initially repurposes a DNN, MANIQA [49], as an extractor for quality-aware features and establishes it as the training backbone for the MA-AGIQA framework. Subsequently, we guide a LMM, mPLUG-Owl2 [50], to focus on fine-grained semantic information through meticulously crafted prompts. We then extract and store the last-layer hidden vector from mPLUG-Owl2, merging it

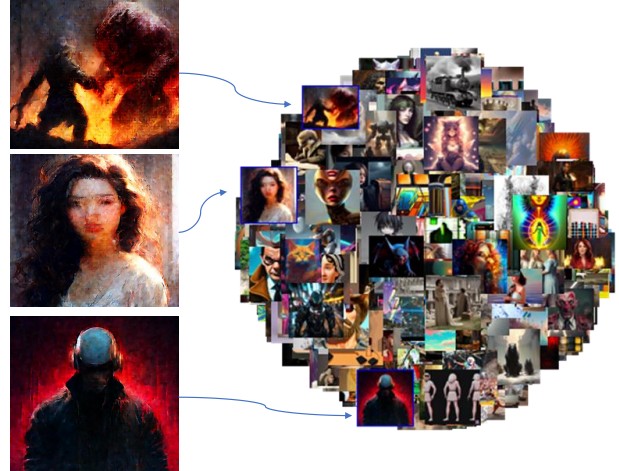

**Figure 2: For the subset of grainy images (extracted from prompts containing "digital" and generated by LCM_Pixart in AIGCQA-20k) that include semantic content, MANIQA achieves an SRCC of 0.2545, which is 70.0% lower than the overall SRCC of 0.8507. In contrast, our MA-AGIQA model achieves an SRCC of 0.8364. It demonstrates that our model possesses a significantly enhanced understanding of AGIs, particularly those whose quality is deeply intertwined with semantic elements.**

with features extracted by MANIQA to infuse the model with rich semantic insights. Finally, we employ a MoE to dynamically integrate quality-aware features with fine-grained semantic features, catering to the unique focal points of different images. As demonstrated in Figure 2, our approach surpasses MANIQA in terms of SRCC, particularly within subsets comprising semantically rich images overflowing with graininess, indicating that our methodology shows remarkable congruence with the human visual system's (HVS) perceptual capabilities. MA-AGIQA achieves SRCC values of 0.8939 and 0.8644 on the AGIQA-3k and AIGCQA-20k datasets, respectively, exceeding the state-of-the-art models by 2.03% and 1.37%, and also demonstrates superior cross-dataset performance.

Our **contributions** are three-fold:

- We systematically analyze the issue of traditional DNN-based IQA lacking the ability to understand the semantic content of AGIs, emphasizing the importance of incorporating semantic information into traditional DNN-based IQA models.
- We introduce the MA-AGIQA model, which incorporates LMM to extract fine-grained semantic features and dynamically integrates these features with traditional DNN-based IQA models.
- We evaluate the MA-AGIQA model on two AI-generated IQA datasets. Experimental results demonstrate that our

model surpasses current state-of-the-art methods without extra training data and also showcases superior cross-dataset performance. Extensive ablation studies further validate the effectiveness of each component.

## 2 RELATED WORK

**Traditional IQA models.** In the field of No-Reference Image Quality Assessment (NR-IQA) [53], traditional models primarily fall into two categories: handcrafted feature-based and DNN-based.

Models based on handcrafted features, such as BRISQUE [24], ILNIQE [54], and NIQE [25], primarily utilize natural scene statistics (NSS) [24, 25] derived from natural images. These models are adept at detecting domain variations introduced by synthetic distortions, including spatial [24, 25, 54], gradient [24], discrete cosine transform (DCT) [36], and wavelet-based distortions [26]. However, despite their effectiveness on datasets with type-specific distortions, these handcrafted feature-based approaches exhibit limited capabilities in modeling real-world distortions.

With the advent of deep learning, CNNs have revolutionized many tasks in computer vision. [13] is pioneer in applying deep convolutional neural networks to NR-IQA. Its methodology employs CNNs to directly learn representations of image quality from raw image patches, bypassing the need for handcrafted features or a reference image. Following this, DBCNN [55] introduces a deep bilinear CNN for blind image quality assessment (BIQA) [53], innovatively merging two CNN streams to address both synthetic and authentic image distortions separately. Furthermore, Hyper-IQA [39], a self-adaptive hyper network, evaluates the quality of authentically distorted images through a novel three-stage process: content understanding, perception rule learning, and quality prediction.

The success of Vision Transformers (ViT) [5] in various computer vision tasks has led to significant advancements. In the realm of IQA, IQT [52] leverages the combination of reference and distorted image features, extracted by CNNs, as inputs for a Transformer-based quality prediction task. MUSIQ [14] utilizes a Transformer to encode distortion image features across three scales, addressing the challenge of varying input image sizes during training and testing. TReS introduces relative ranking and self-consistency loss to capitalize on the abundant self-supervisory information available, aiming to decrease the network's sensitivity. What's more, MANIQA [49] explored multi-dimensional feature interaction, utilizing spatial and channel structural information to calculate a non-local representation of the image, enhancing the model's ability to assess image quality comprehensively.

**LMMs for IQA.** Recent methodologies employing LMMs for IQA either utilize LMMs in isolation or combine them with DNNs as feature extractors to enhance performance. [30] introduces an innovative image-prompt fusion module, along with a specially designed quality assessment token, aiming to learn comprehensive representations for AGIs, providing insights from image-prompt alignment. However, the evaluation of AGIs in practical scenarios often does not involve prompts and image-prompt alignment is more significant for assessing the capabilities of generative models rather than images quality. CLIPIQA [42] signifies a breakthrough in assessing image quality and perception by harnessing the strengths of CLIP [31] models. This method bridges the divide between measurable image quality attributes and subjective perceptions of quality without necessitating extensive labeling efforts. Nonetheless, their [30, 42] dependence on visual-text similarity for quality score prediction often constrains their performance, rendering it marginally less effective compared to methods that exclusively focus on visual analysis. What's more, Q-Bench [43] innovates with a softmax strategy, allowing LMMs to deduce quantifiable quality scores. This is achieved by extracting results from softmax pooling on logits corresponding to five quality-related tokens. And Q-Align [45] employs strategic alignment techniques to foster accuracy. Expanding further, [47] delves into enhancing the assessment of AGIs by focusing on optimizing individual text prompts to leverage the intrinsic capabilities of LMMs, aiming to provide a more nuanced understanding and evaluation of image quality of AGIs. However, these methods, while notable, fall short of achieving satisfying efficacy, leaving considerable room for improvement.

## 3 METHOD

As depicted in Figure 3, framework of MA-AGIQA is structured into three sections. Section 3.1 introduces our adoption of a DNN, specifically MANIQA [49], tailored for the AGIs quality assessment task, serving as our primary training backbone. In Section 3.2, we incorporate the LMM mPLUG-Owl2 [50] as a feature extractor. This component is crucial for acquiring fine-grained semantic features via carefully crafted text prompts. Lastly, Section 3.3 addresses the variability in focal points across different images. To adaptively integrate the feature vectors during training, we utilizes a MoE structure for feature fusion. This approach ensures that the most salient features are emphasized. Further details are elaborated below.

### 3.1 Quality-aware Feature Extraction

To leverage the capability of DNNs to adapt to the data distribution of specific tasks, we employ MANIQA [49] as a quality-aware feature extractor. MANIQA enhances the evaluation of image quality by applying attention mechanisms across both the channel and spatial dimensions, thereby increasing the interaction among various regions of the image, both globally and locally. This approach generates projections $weight$ ($W$) and $score$ ($S$) for a given image, and the final rating of the whole image is determined through the sum of multiplication of $S$ by $W$, which can be illustrated as Equation (1) :

$$(S, W) = \mathcal{T}([image]),$$
$$rating = \frac{\sum S \times W}{\sum W}, \tag{1}$$

where $S$ and $W$ are one dimensional vectors.

However, directly applying MANIQA to the quality assessment of AGIs presents challenges, as illustrated in Figure 4. Image (a) displays a complex, symmetrical pattern, devoid of meaningful semantic content. Image (b) features incoherent areas, such as two grey holes in the sky that are inconsistent with the common sense. The blurriness and fuzziness of the man's face in image (c) along the edges significantly impair human perception. Conversely, image (d), despite its severe graininess, retains its semantic integrity,

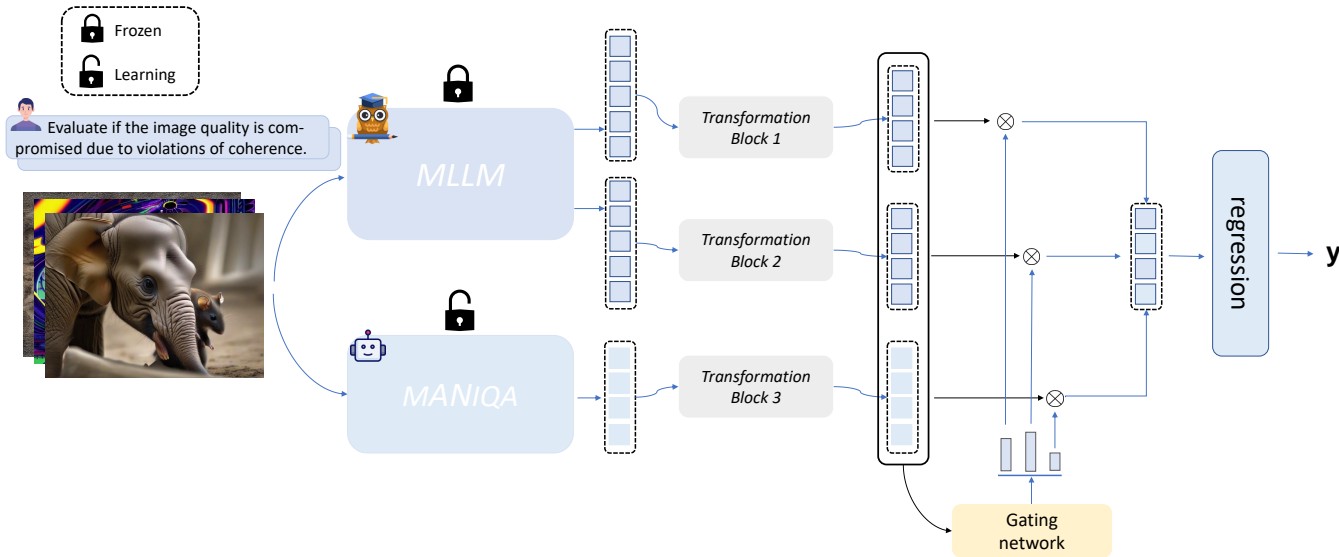

Figure 3: Overview of our proposed MA-AGIQA framework. Initially, MANIQA is repurposed as the foundational training backbone, whose structure is modified to generate quality-aware features. Second, a parameter fixed LMM, mPLUG-Owl2, serves as a fine-grained semantic feature extractor. This module utilizes carefully crafted prompts to capture the desired semantic information. Finally, the AFM module acts as an organic feature integrator, dynamically combining these features for enhanced performance.

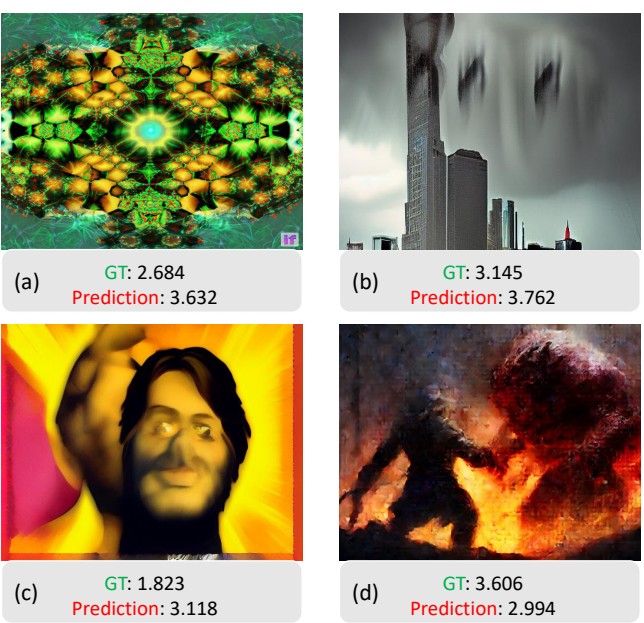

| (a) | GT: 2.684 Prediction: 3.632 |
| (b) | GT: 3.145 Prediction: 3.762 |
| (c) | GT: 1.823 Prediction: 3.118 |
| (d) | GT: 3.606 Prediction: 2.994 |

Figure 4: Four types of image display with strong correlation between image quality and semantics. The ground truth and model predication of the relevant images are presented below each image, showing a significant difference between the model predication and the ground truth, indicating that the model's understanding of semantics is not sufficient.

representing an appealing artistic form. Traditional DNN-based models like MANIQA, lacking the capacity to comprehend semantic content, tend to overestimate the quality of images (a), (b), and (c), resulting in scores much higher than the ground truth. However, these images should be rated as low quality due to the poor viewing experience they offer. For image (d), traditional DNN-based models focus excessively on the graininess, mistaking it for a flaw, and assign a score significantly lower than the ground truth. This highlights the critical need for incorporating semantic information into the quality assessment of AGIs by traditional DNN-based models.

To address this issue, modifications were made so that the generated $S$ and $W$ no longer produce a rating. Instead, they yield a quality-aware feature $f_1$, setting the stage for the subsequent fusion with features extracted by LMM. $f_1$ is generated as:

$$f_1 = S \times W. \tag{2}$$

During the training phase, the parameters of modified MANIQA are continuously updated. This refinement process ensures that MANIQA can extract features more relevant to the quality of AGIs. Furthermore, the training process facilitates a more seamless integration between MANIQA and LMM, leading to superior outcomes.

## 3.2 Fine-grained Semantic Feature Extraction

LMMs are capable of understanding and analyzing the semantic content of images and their relationship with human cognition. They assess whether different parts of an image form a cohesive whole and evaluate whether the elements within the picture are semantically coherent [7, 21, 28]. mPLUG-Owl2 [50] employs a modality-adaptive language decoder to handle different modalities

within distinct modules, which mitigates the issue of modality interference. Given the importance of effectively guiding the model through textual prompts to elicit the desired output, we have selected mPLUG-Owl2 as our feature extractor.

We consider the application of mPLUG-Owl2 in the following aspects of semantic content:

- **Existence of Semantic Content.** The importance of semantic content in an image lies in its ability to convey a clear and meaningful message to the viewer. An image lacking in semantic content may be difficult to understand, fail to effectively convey its intended message, reducing audience engagement and satisfaction.
- **Coherence of Semantic Content.** The coherence of semantic content in an image relates to whether the generated image can provide a coherent, logically sound visual experience for human viewers. When the various parts of an image are semantically consistent, it is better able to convey a clear story, emotion, or message. In contrast, any inconsistency in the primary focus of images will greatly detract from their quality and convey a significantly negative impression.

Consequently, we try to propose the rational design of prompts leading LMMs to obtain those image semantic content. mPLUG-Owl2 possess the ability to understand fine-grained semantic contents, but without carefully designed input prompts, some prompts, such as *"Please evaluate if the image quality is compromised due to violations of common human sense or logic?"* although it expresses the desire for the model to assess whether the semantic content of the image contradicts human perception, would lead to unsatisfactory results. To better utilize mPLUG-Owl2 for the task of evaluating AGIs, we meticulously designed prompts to guide the LMM. Specifically, we designed two prompts, denoted as $prompt_a$ and $prompt_b$ respctively,

- "Evaluate the input image to determine if its quality is compromised due to a lack of meaningful semantic content."
- "Evaluate if the image quality is compromised due to violations of coherence."

corresponding to the existence of semantic content and the coherence of semantic content in images, respectively. Test results, as shown in Figure 5 using the mPLUG-Owl2 official demo[1], have proven these questions to be effective.

However, the textual output from mPLUG-Owl2 is not immediately conducive to being utilized by MANIQA to impart semantic insights. To bridge this gap, it's essential to obtain the information provided by mPLUG-Owl2 into a format that MANIQA can easily leverage. SO we extract features from the final layer of mPLUG-Owl2's hidden layers, achieving an accessible embedded representation of the LMM's output. This output is a tensor with dimensions of [token_length, hidden_size], where "token_length" represents the number of output tokens, and "hidden_size" denotes the dimensionality of the hidden layer representations associated with each token. For mPLUG-Owl2, the hidden_size is set to 4096. Subsequently, we conduct an averaging operation across the token dimension, yielding a vector with dimensions 1x4096. This vector

[1]https://modelscope.cn/studios/iic/mPLUG-Owl2/summary

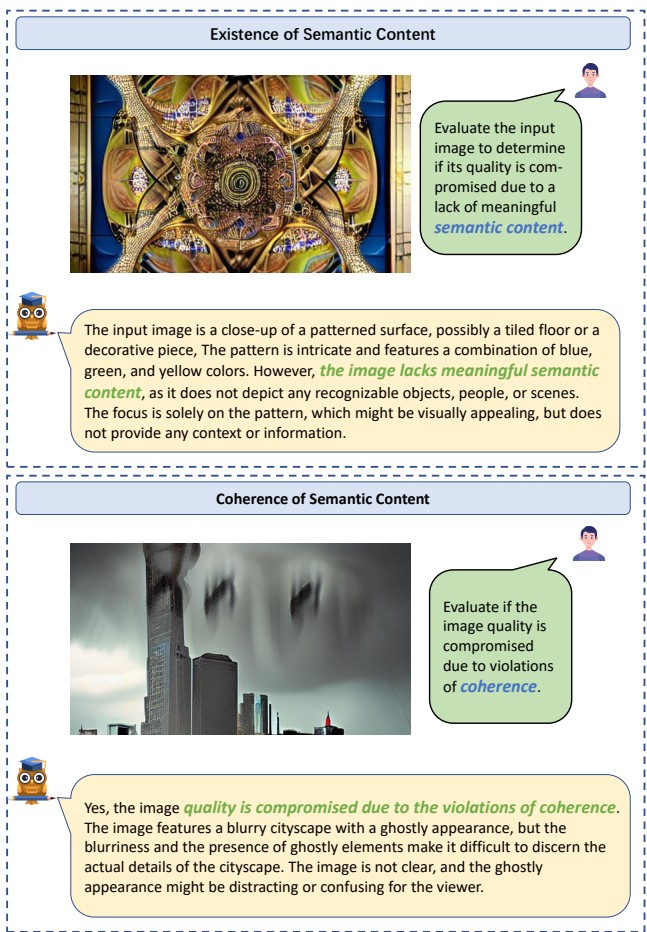

**Figure 5: Presentation of mPLUG-Owl2's answers to two prompts.**

then serves as the basis for further feature fusion procedures. The process can be represented as Equation (3) :

$$(\mathbf{m}_i^1, \mathbf{m}_i^2, \cdots, \mathbf{m}_i^n) = \mathcal{M}([image], [prompt_i])[-1],$$
$$f_i = Average(\mathbf{m}_i^1, \mathbf{m}_i^2, \cdot, \mathbf{m}_i^n), \text{ where } i \in \{a, b\}, \tag{3}$$

where $\mathbf{m}_i^k$ represents a hidden vector of token $k$ corresponding to $prompt_i$, and $\mathcal{M}$ denotes mPLUG-Owl2.

It is important to note that throughout the entire training and testing process, the parameters of mPLUG-Owl2 are fixed. Because mPLUG-Owl2 is typically pre-trained on large-scale datasets and has already learned a rich set of joint visual and language knowledge, it can effectively capture the fine-grained semantic information relevant to input prompts, even with fixed parameters. Additionally, fine-tuning LMMs in every training iteration would significantly increase training time. Using it solely as a feature extractor significantly reduces computational costs, making the training process more efficient. So, we pre-obtain and save the semantic content features of each image in advance.

## 3.3 Adaptive Fusion Module

Given the complex influence of color, composition, details, semantic content, and other factors on image quality, simply concatenating the extracted features may not always yield the best results. To dynamically fuse a variety of complementary features, we propose the adaptive fusion module (AFM) for organic feature integration. This process can be divided into two main parts. **The first part** involves transforming the extracted features into a unified vector space of the same dimension, allowing for vector fusion operations. Specifically, for features extracted by MANIQA, this transformation block applies a fully connected (Fc) layer, transforming them to the same dimension as the original features (1x784) to provide a richer combination. For features derived from mPLUG-Owl2, it uses a Fc layer to project them onto a 1x784 dimension, followed by a relu activation layer and a dropout layer to enhance the network's expressive power and generalization. **The second part** employs a MoE to dynamically fuse the three features. The MoE's gating network takes the transformed three features as input and outputs dynamic weights $\alpha$, corresponding to the three features' contributions to image quality. Structurally, this gating network comprises a Fc layer and a sigmoid layer. The final image quality representation vector $g$ can be obtained through a weighted sum of the three feature vectors. Following the denotation which sign the three features as $f_1$, $f_a$, $f_b$, this process can be represented as:

$$
\begin{aligned}
f_i' &= \mathcal{F}_i^{trans}(f_i), \text{ where } f_i' \in \mathbb{R}^d, \\
\alpha &= \mathcal{F}^{gate}(\text{Concat}(f_1', f_a', f_b')), \text{ where } \alpha \in \mathbb{R}^3, \\
g &= \sum_{i=1}^{3} f_i' \cdot \alpha_i, \text{ where } g \in \mathbb{R}^d, \ i \in \{1, a, b\},
\end{aligned}
\tag{4}
$$

where $\mathcal{F}_i^{trans}$ is the transformation block of feature $i$, and $\mathcal{F}^{gate}$ is the gating network's mapping function, $f_i$ is the original extracted feature and $f_i'$ is the transformed feature. $\mathbb{R}^d$ is the dimension space of $f_i'$. Finally, we obtain the final image quality score output through a simple regression layer, consisting of a Fc layer.

## 4 EXPERIMENTS

### 4.1 Dataset and Evaluation Metrics

**Dataset.** Our model is evaluated on two AI-Generated image datasets, including AIGCQA-20k [17] and AGIQA-3k [20]. Specifically, AIGCQA-20k contains 20k images, but at the time of writing, only 14k images have been published. Our experiments are conducted on these 14k images. The MOS for AIGCQA-20k images are distributed between 0-5, with higher scores indicating better image quality. Images in AIGCQA-20k are generated by 15 models, including DALLE2 [32], DALLE3 [32], Dream [6], IF [4], LCM Pixart [22], LCM SD1.5 [22], LCM SDXL [22], Midjourney [11], Pixart $\alpha$ [3], Playground [29], SD1.4 [34], SD1.5 [34], SDXL [35] and SSD1B [8]. AGIQA-3k includes 2982 images, with MOS also distributed between 0-5, where higher values represent better quality. Images in AGIQA-3k are derived from six models, including GLIDE [27], Stable Diffusion V-1.5 [34], Stable Diffusion XL-2.2 [35], Midjourney [11], AttnGAN [48], and DALLE2 [32]. During training, we split the entire dataset into 70% for training, 10% for validation, and 20% for testing. To ensure the same set of images in each subset

when testing across different models, we set the same random seed during the split to control variables and ensure reproducibility.

**Evaluation Metric.** Spearman's Rank-Order Correlation Coefficient (SRCC), Pearson's Linear Correlation Coefficient (PLCC), the Kullback-Leibler Correlation Coefficient (KLCC), and the Root Mean Square Error (RMSE) are selected as metrics to measure monotonicity and accuracy. SRCC, PLCC, and KLCC range from -1.0 to 1.0, with larger values indicating better results. In our experiments, we employ the sum of SRCC and PLCC as the criterion for selecting the optimal validation case, and emphasize SRCC for comparing model performance.

### 4.2 Implementation Details

Our method is implemented based on PyTorch, and all experiments are conducted on 4 NVIDIA 3090 GPUs. For all datasets, we opt for handcrafted feature-based BRISQUE [24], NIQE [25] and IL-NIQE [54], deep learning (DL)-based HyperIQA [39], MANIQA [49], MUSIQ [14], DBCNN [55], StairIQA [40], BAID [51], and LMM-based CLIPIQA [42], CLIPIQA+ [42] and Q-Align [45]. During the training process of deep learning models, we use the Adam optimizer [15] with a weight decay of 1e-5, and the initial learning rate is 1e-5. The batch size is 8 during training, validation, and testing. All DL-based models are trained for 30 epochs using MSE loss and validated after each training process. The checkpoint with the highest sum of SRCC and PLCC during validation is used for testing. Handcrafted feature-based and LMM based models are used directly without training.

### 4.3 Comparison with SOTA methods

Table 1 lists the results of MA-AGIQA and 12 other models on the AGIQA-3k and AIGCQA-20k dataset. It has been observed that LMM-based models significantly outperform those that rely on handcrafted features. This superior performance is attributed to LMMs being trained on extensive datasets, which provides them with a robust understanding of images and enhances their generalizability. However, trained DL-based models generally perform far better than the LMM-based models because DL-based models tend to fit the data distribution of specific tasks better, thereby resulting in improved performance. Among these twelve models, the ViT-based MANIQA outperforms the other eleven models, and our method still significantly surpasses it on the same training and testing split with large margins (+3.72% of SRCC, +1.73% of PLCC and +5.43% of KRCC in AGIQA-3k & +1.61% of SRCC, +2.02% of PLCC and +2.90% of KRCC in AIGCQA-20k). This demonstrates the superiority of integrating features extracted by LMM into traditional DNN, significantly improving the accuracy and consistency of prediction results.

To evaluate the generalization capability of our MA-AGIQA model, we conducted cross-dataset evaluations. Table 2 shows that MA-AGIQA significantly outperforms the other two models, Hyper-IQA and StairIQA, which performed best on single datasets, with large margins. This superior performance can largely be attributed to the robust generalization capability of the LMM and the benefits of the MoE architecture, which excels in dynamically fusing features.

**Table 1: Comparisons with SOTA (State-Of-The-Art) methods on AGIQA-3k and AIGCQA-20K-Image datasets. The up arrow "↑" means that a larger value indicates better performance. The best and second best performances are bolded and underlined, respectively. MA-AGIQA outperforms existing SOTA methods on both datasets by large margins. Note: to ensure fair comparisons, we trained and tested all deep learning based models and ours with the same dataset splitting method.**

| Type | Method | AGIQA-3k | | | | AIGCQA-20K-Image | | | |
|---|---|---|---|---|---|---|---|---|---|
| | | SRCC↑ | PLCC↑ | KRCC↑ | RMSE↓ | SRCC↑ | PLCC↑ | KRCC↑ | RMSE↓ |
| Handcrafted feature-based | BRISQUE [24] | 0.4726 | 0.5612 | 0.3227 | 0.8299 | 0.1663 | 0.3580 | 0.1112 | 0.6813 |
| | NIQE [25] | 0.5236 | 0.5668 | 0.3637 | 0.8260 | 0.2085 | 0.3378 | 0.1394 | 0.6868 |
| | ILNIQE [54] | 0.6097 | 0.6551 | 0.4318 | 0.7576 | 0.3359 | 0.4551 | 0.2290 | 0.6497 |
| LMM-based | CLIPIQA [42] | 0.6524 | 0.6968 | 0.4632 | 0.7191 | 0.4147 | 0.6459 | 0.2861 | 0.5570 |
| | CLIPIQA+ [42] | 0.6933 | 0.7493 | 0.4957 | 0.664 | 0.4553 | 0.6682 | 0.3169 | 0.5428 |
| | Q-Align [45] | 0.6728 | 0.6910 | 0.4728 | 0.7204 | 0.6743 | 0.6815 | 0.4808 | 0.5199 |
| Traditional DNN-based | HyperIQA [39] | 0.8509 | 0.9049 | 0.6685 | 0.4134 | 0.8162 | 0.8329 | 0.6207 | 0.3902 |
| | MANIQA [49] | 0.8618 | 0.9115 | 0.6839 | 0.4111 | 0.8507 | 0.8870 | 0.6612 | 0.3273 |
| | DBCNN [55] | 0.8263 | 0.8900 | 0.6393 | 0.4533 | 0.8054 | 0.8483 | 0.6121 | 0.3726 |
| | StairIQA [40] | 0.8343 | 0.8933 | 0.6485 | 0.4510 | 0.7899 | 0.8428 | 0.6053 | 0.3927 |
| | BAID [51] | 0.1304 | 0.2030 | 0.0854 | 0.9487 | 0.1652 | 0.1483 | 0.1279 | 0.7297 |
| | MUSIQ [14] | 0.8261 | 0.8657 | 0.6400 | 0.4907 | 0.8329 | 0.8646 | 0.6403 | 0.3634 |
| DL with LMM | **MA-AGIQA** | **0.8939** | **0.9273** | **0.7211** | **0.3756** | **0.8644** | **0.9050** | **0.6804** | **0.3104** |

**Table 2: Cross-dataset performance comparison for M-AIGQ-QA, HyperIQA, and StairIQA. "Direction" from A to B means training with train subset of dataset A and testing on test subset of dataset B. The best result is bolded.**

| | direction | SRCC ↑ | PLCC ↑ | KRCC ↑ | RMSE ↓ |
|---|---|---|---|---|---|
| MA-AGIQA | 20k→3k | **0.8053** | **0.8430** | **0.6083** | **0.5399** |
| | 3k→20k | **0.7722** | **0.8314** | **0.5777** | **0.4055** |
| HyperIQA | 20k→3k | 0.6820 | 0.6806 | 0.4806 | 0.7352 |
| | 3k→20k | 0.6374 | 0.6547 | 0.4577 | 0.5414 |
| StairIQA | 20k→3k | 0.4335 | 0.5234 | 0.3294 | 0.8549 |
| | 3k→20k | 0.6495 | 0.6895 | 0.4644 | 0.5285 |

**Table 3: Ablation studies of different component combinations in the MA-AGIQA model on AGIQA-3k. SRCC, PLCC and KRCC are reported. The best result is bolded. Note: "semantic feature" and "coherence feature" denote features extracted by mPLUG-Owl2 through $prompt_a$ and $prompt_b$ respectively.**

| MANIQA | Semantic Feature | Coherence Feature | SRCC↑ | PLCC↑ | KRCC↑ |
|---|---|---|---|---|---|
| ✓ | | | 0.8800 | 0.9196 | 0.7031 |
| | ✓ | | 0.8662 | 0.9082 | 0.6823 |
| | | ✓ | 0.8661 | 0.9084 | 0.6821 |
| ✓ | ✓ | | 0.8685 | 0.9108 | 0.6853 |
| ✓ | | ✓ | 0.8820 | 0.9197 | 0.7090 |
| | ✓ | ✓ | 0.8699 | 0.9102 | 0.6867 |
| ✓ | ✓ | ✓ | **0.8939** | **0.9273** | **0.7211** |

## 4.4 Ablation Study

**Necessity of Fine-grained Semantic Features.** To assess the benefits of integrating features extracted by mPLUG-Owl2 [50] into MANIQA [49], we carried out comprehensive ablation studies on each component and their various combinations, as detailed in Tables 3 and 4. Our findings indicate that using either the features extracted by the LMM alone or solely relying on a traditional network does not yield the best outcomes. In contrast, integrating one fine-grained semantic feature with the original MANIQA network can enhance the network's performance. However, the optimal results were achieved by combining two features extracted by the LMM with MANIQA, which led to significant improvements on the AGIQA-3k dataset (increases of 1.57%, 0.83%, and 2.56% in SRCC, PLCC, and KRCC, respectively) and on the AIGCQA-20k dataset (enhancements of 2.72%, 1.94%, and 4.35%).

The marked enhancements achieved by incorporating two fine-grained semantic features suggest that LMM is adept at capturing nuanced, complex features that traditional models might overlook, fostering a more thorough understanding and assessment of AGIs quality. The results from these ablation experiments highlight the significant contribution of fine-grained semantic features.

**Contribution of MoE.** Table 5 demonstrates that incorporating the MoE structure, rather than simply concatenating three vectors, does indeed improve network performance, albeit marginally. Specifically, on the AGIQA-3k dataset, we observed increases of 0.20%, 0.17%, and 0.16% in SRCC, PLCC, and KRCC, respectively. For the AIGCQA-20k dataset, the improvements were 0.67%, 0.95%, and 1.37%. The gains, although seemingly modest, highlight the potential of MoE structure in complex systems where integrating diverse expertise can yield better decision-making and predictive outcomes.

**Table 4: Ablation studies of different component combinations in the MA-AGIQA model on AIGCQA-20k. SRCC, PLCC and KRCC are reported. The best result is bolded. Note: "semantic feature" and "coherence feature" denote features extracted by mPLUG-Owl2 through $prompt_a$ and $prompt_b$ respectively.**

| MANIQA | Semantic Feature | Coherence Feature | SRCC↑ | PLCC↑ | KRCC↑ |
|--------|------------------|-------------------|-------|-------|-------|
| ✓ |  |  | 0.8415 | 0.8877 | 0.6520 |
|  | ✓ |  | 0.8184 | 0.8345 | 0.6323 |
|  |  | ✓ | 0.8181 | 0.8343 | 0.6320 |
| ✓ | ✓ |  | 0.8540 | 0.8975 | 0.6671 |
| ✓ |  | ✓ | 0.8596 | 0.9016 | 0.6738 |
|  | ✓ | ✓ | 0.8180 | 0.8323 | 0.6312 |
| ✓ | ✓ | ✓ | **0.8644** | **0.9050** | **0.6804** |

**Table 5: Ablation studies on the MoE structure in the AFM demonstrate that compositions integrating MoE yield superior results on both AGIQA-3k and AIGCQA-20k datasets. The better result is bolded.**

| dataset | MoE | SRCC ↑ | PLCC ↑ | KRCC ↑ | RMSE ↓ |
|---------|-----|--------|--------|--------|--------|
| 3k | ✗ | 0.8921 | 0.9257 | 0.7199 | 0.3797 |
|  | ✓ | **0.8939** | **0.9273** | **0.7211** | **0.3756** |
| 20k | ✗ | 0.8586 | 0.8964 | 0.6712 | 0.3234 |
|  | ✓ | **0.8644** | **0.9050** | **0.6804** | **0.3104** |

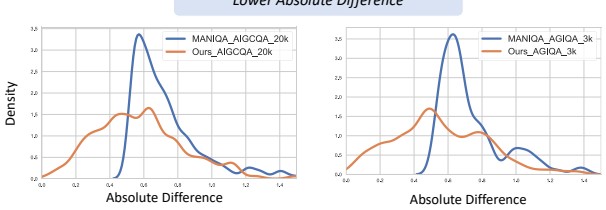

**Figure 6: Comparative Density Distributions of Absolute Differences for MANQA and MA-AGIQA on AGIQA-3k and AIGQA-20k Datasets**

### 4.5 Visualization

To vividly demonstrate the efficacy of the MA-AGIQA framework, we selected 300 images from the AIGCQA-20k and AGIQA-3k datasets where MANIQA had the poorest performance. These images primarily exhibit issues in semantic content. We computed the absolute values of the differences between the model scores and the image ground truth, and illustrated these differences in Figure 6, using 0.1 as the bin size for plotting the quality score distribution. The results clearly show that our MA-AGIQA model are more closely aligned with human perception, with a noticeable shift

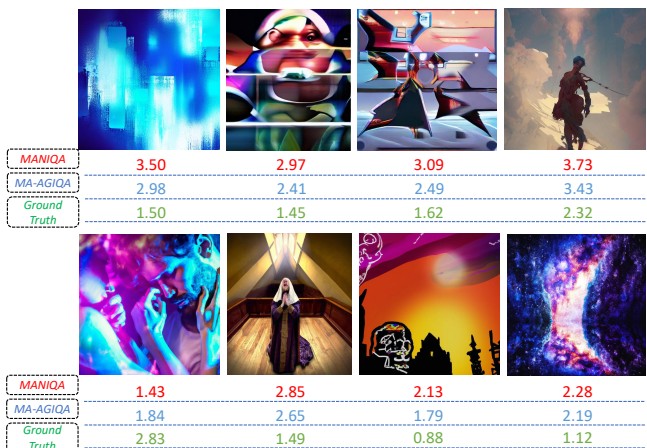

**Figure 7: Comparative Analysis of Image Quality Assessment Models: Evaluating MANIQA versus MA-AGIQA Against Ground Truth Scores**

in the difference distribution toward zero and a marked reduction in peak values.

Figure 7 presents a collection of images where the assessments from the MANIQA model were mostly off the mark. Scores assigned by MANIQA alongside those given by the proposed MA-AGIQA model and the ground truth are listed, which reveal that the MA-AGIQA model markedly enhances alignment with the ground truth in contrast to MANIQA. For instance, in the first image of the top row, MANIQA's score is 3.50, which diverging substantially from the ground truth score of 1.50. However, MA-AGIQA's score is 2.98, demonstrating a much closer approximation to the ground truth. This pattern is consistent across the images shown, with MA-AGIQA consistently producing scores that are closer to the ground truth, reflecting a more accurate assessment of image quality.

## 5 CONCLUSION

To mitigate the shortcomings of traditional DNNs in capturing semantic content in AGIs, this study explored the integration of LMMs with traditional DNNs and introduced the MA-AGIQA network. Leveraging mPLUG-Owl2 [50], our network efficiently extracts semantic features to enhance MANIQA [49] for quality assessment. The MA-AGIQA network's ability to dynamically integrate fine-grained semantic features with quality-aware features enables it to effectively handle the varied quality aspects of AGIs. Experiment results across two prominent AGIs datasets confirm our model's superior performance. Through thorough ablation studies, the indispensable role of each component within our framework has been validated. This research aspires to catalyze further exploration into the fusion of LMMs within AI-generated content quality assessment and envisions broader application potentials for such methodology.

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
