# OpenReview forum: "Large Multi-modality Model Assisted AI-Generated Image Quality Assessment"
_acmmm.org/ACMMM/2024/Conference — MM2024 Oral_

### Official Review · Reviewer_wUPA · 2024-05-12

**Rating:** 6
**Confidence:** 4

**Summary:**

This paper addresses the challenge of assessing the quality of AI-generated images (AGIs), which traditional deep neural network (DNN) models struggle with due to their inability to handle semantic inaccuracies typical of AGIs. To overcome these limitations, the authors introduce a novel model named MA-AGIQA that integrates large multi-modality models (LMMs) with traditional DNNs to enhance the assessment of semantic content within images. The paper's achievements are demonstrated through comprehensive experiments, where MA-AGIQA outperforms existing state-of-the-art models by a large margin, showcasing improved generalization capabilities and a deeper understanding of the quality-related aspects of AGIs.

**Strengths:**

1.Identifying the limitations of traditional DNN-based IQA models in assessing AGIs, particularly their inability to recognize and interpret semantic content.
2.Introducing an innovative approach, the MA-AGIQA model that incorporates LMMs for extracting fine-grained semantic features and dynamically integrates these with traditional DNN features to improve quality assessment.
3.Demonstrating through extensive experiments on two AI-generated images datasets that MA-AGIQA achieves state-of-the-art performance, significantly improving the evaluation of image quality in AGIs compared to existing models.
4.Opening future research directions that integrate LMMs for semantic analysis in image quality and encouraging further exploration into how these models can be optimized and adapted for other types of multimedia content beyond AI-generated images.
5. The paper's focus on improving image quality assessment for AI-generated images is highly relevant to the ACM MM community, including aspects of visual media analysis, semantic content interpretation, and the integration of multimodal data for enhanced computational understanding.

**Limitations:**

There are some minor problems, which need to be solved before publication. If the following problems are well-addressed, this reviewer believes that the essential contribution of this paper is important for IQA.

1.Ensure that appropriate references are added at relevant sections of the paper, such as in the 'Related Work' where TReS is discussed.
2.Carefully review the text for any grammatical and spelling errors to maintain the professionalism and readability of the paper.

**Suitability:**

3

---

### Official Review · Reviewer_TjxM · 2024-05-27

**Rating:** 4
**Confidence:** 3

**Summary:**

This paper proposes a quality assessment method for AI-generated images, which combines a visual language model with an IQA model. Experimental results show that the method achieves good performance.

**Strengths:**

The method is designed with prompts for judging semantic information, thus solving the problem that the model has difficulty in extracting semantic information of AI-generated images. Specifically, coarse-grained semantic features are extracted using a visual language model and fine-grained distorted features are extracted using a traditional IQA model. This approach combines the advantages of both to achieve excellent results.

**Limitations:**

1. Cross-database experiments should be compared with AGIQA method instead of traditional IQA model.
2. There are fewer experiments in the paper. Performance comparison of traditional IQA benchmark databases can be added.

**Suitability:**

2

---

### Official Review · Reviewer_ffSf · 2024-06-02

**Rating:** 4
**Confidence:** 3

**Summary:**

Compared to general (real) image quality assessment, AI-generated image quality assessment is more complex due to unexpected components (contents) in images. The authors show that existing methods of IQA on AGI datasets have weaknesses and analyze them. They apply LMM models to address this problem and achieve high performance compared to existing methods.

**Strengths:**

[S1] The proposed method is novel with logical grounds. They analyze why the traditional DNN-based models cannot overcome the performance that the models can achieve in real-image quality assessment. Based on the analysis, they conclude that using LMM models is beneficial to IQA on AGI datasets.

[S2] The paper is well-written and easy to understand.

**Limitations:**

[W1] The traditional DNN-based methods might have weaknesses when it comes to AGI, and therefore, introducing LMM seems like a good idea. However, the analysis of these weaknesses is not convincing. The evidence presented relies solely on Fig. 4, which does not adequately represent the overall trends in the AGI dataset. The authors need to demonstrate that this is a universally observable trend. For instance, they could show how many mis-predicted samples among N sampled images are distributed across a, b, c, and d in Fig. 4.

[W2] (minor) Using LMM in conjunction with traditional methods is likely to require more computational resources, and this should be reported when comparing performance.

[W3] (minor) Evaluating the quality of generated images is a crucial part of developing generative AI models. While it is common to evaluate the generated image dataset as a whole, recent techniques have been developed to assess individual generated images[1,2,3]. It would be beneficial to discuss how IQA on AGIs dataset fundamentally operates differently from these methods.

[1] Improved precision and recall metric for assessing generative models, NeurIPS 2019
[2] Rarity score : A new metric to evaluate the uncommonness of synthesized images, ICLR 2023
[3] Anomaly Score: Evaluating Generative Models and Individual Generated Images based on Complexity and Vulnerability, CVPR 2024

**Suitability:**

3

---

### Meta-Review · Area_Chair_UzGm · 2024-06-30

**Recommendation:** Accept (Oral)
**Confidence:** 4

**Metareview:**

The final ratings are 2 weak accept and 1 accept. The reviewers acknowledged novelty and good performance. Initial questions were addressed through the rebuttal.